# The Impact of Rice–Frog Co-Cultivation on Greenhouse Gas Emissions of Reclaimed Paddy Fields

**DOI:** 10.3390/biology14070861

**Published:** 2025-07-16

**Authors:** Haochen Huang, Zhigang Wang, Yunshuang Ma, Piao Zhu, Xinhao Zhang, Hao Chen, Han Li, Rongquan Zheng

**Affiliations:** 1Provincial Key Laboratory of Wildlife Biotechnology and Conservation and Utilization, Zhejiang Normal University, Jinhua 321004, China; 19852999776@163.com (H.H.); mys000408@zjnu.edu.cn (Y.M.); zhupiao36@zjnu.edu.cn (P.Z.); star_233@163.com (X.Z.); zij0900@163.com (H.C.); lihan010303@zjnu.edu.cn (H.L.); 2Department of Basic Medicine, College of Medicine, Jinhua University of Vocational Technology, Jinhua 321017, China; 15906896231@163.com; 3Xingzhi College, Zhejiang Normal University, Jinhua 321004, China

**Keywords:** greenhouse gas mitigation, reclaimed paddy field, methane, nitrous oxide

## Abstract

Reclaimed paddy fields, with low soil fertility and suboptimal biological conditions, face challenges in sustainable agricultural production. This study evaluated the effects of rice–frog co-cultivation at different frog densities (low: LRF; high: HRF) on greenhouse gas (GHG) emissions compared to a rice monoculture (CG). Results showed that co-cultivation significantly reduced cumulative CH_4_ emissions by 34.8% (LRF) and 48.0% (HRF), driven by the altered soil pH, cation exchange capacity (CEC), and shifts in methanogen (*mcrA*) and methanotroph (*pmoA*) gene abundances. Conversely, N_2_O emissions increased by 37.1% (LRF) and 96.3% (HRF), linked to the elevated soil redox potential (Eh), urease activity, and denitrifier gene (nirS, nirK) abundances. Due to CH_4_’s dominant contribution (93.1–98.2%) to the global warming potential (GWP), co-cultivation reduced the overall GWP by 33.5% (LRF) and 45.3% (HRF), with high-density co-cultivation being the most effective, highlighting its potential as a sustainable strategy for reclaimed paddy fields.

## 1. Introduction

As the global population has continued to increase, ensuring food production has become the top priority for food security in various countries [1]. Thus, there is a particularly urgent need to utilize currently abandoned farmland and resume its utilization in food production. Land reclamation has a positive social significance for the intensive utilization of idle land and the promotion of agricultural industry development [2]. However, compared with conventional cultivated land, reclaimed fields have some critical deficiencies. For example, they have lower soil fertility, poorer soil physicochemical properties and crop adaptability, and less favorable soil biological conditions. Their productivity is thus far lower than that of typical arable land [3,4,5,6]. Therefore, determining how to quickly improve the fertility of reclaimed soil and optimize its physicochemical properties and biological conditions has become a key objective in ecological research on reclaimed cultivated land.

Rice is the main staple crop for approximately half of the global population. However, paddy fields, owing to their unique anaerobic environment, have become a substantial source of both CH_4_ and N_2_O, major greenhouse gasses that have a great impact on global warming [7,8,9]. Against the backdrop of the intensifying global warming and ever-increasing demand for food, one crucial agricultural research goal is the development of approaches to reduce greenhouse gas emissions from paddy fields while maintaining stable and high rice yields [10,11]. To this end, modern agriculture is exploring more eco-agricultural systems. Integrated rice–aquaculture is an efficient agricultural model with a virtuous ecological cycle [12]. It improves the utilization efficiency of land resources, reduces negative environmental impacts, increases farmers’ incomes, and promotes sustainable agricultural development. Owing to its advantages, such as enhancing rice yields, improving paddy field soil quality, and increasing economic returns, this model has gradually gained popularity [13]. The various activities of different aquatic animals in the paddy field ecosystem, such as movement and feeding, can alter the paddy field ecological environment [14]. Long-term studies in multiple geographic regions have confirmed that rice–duck co-cultivation systems significantly mitigate CH_4_ emissions in paddy fields. During the rice growth cycle, the activities of ducks can reduce CH_4_ emissions by 8.72–14.18% through mechanisms such as loosening the soil layer and improving the soil microflora. The activities of ducks in paddy fields increase the contact between the soil and oxygen, enhance the activity of methane-oxidizing bacteria, and promote the oxidation of CH_4_ produced in the soil by methane-oxidizing bacteria, thus reducing overall CH_4_ emissions [15,16]. In rice–shrimp co-cultivation systems, shrimps increase the oxygen concentration at the soil–water interface through various activities, such as nest-building and foraging, and regulate the abundance of the soil microflora expression, resulting in a reduction in total CH_4_ emissions in paddy fields. This decrease is caused by the abundance of anaerobic methanogens decreasing, while the abundance of methane–oxidizing bacteria was observed to increase significantly [17]. However, in rice–fish co-cultivation systems, the respiration of fish consumes oxygen, thus reducing the oxidation–reduction potential and oxygen content in the soil and enhancing the activity of methanogens, which leads to the increased CH_4_ emissions of paddy fields [16,18,19,20].

The rice–frog co-cultivation model utilizes the natural symbiotic relationship between frogs and rice to increase rice yields, improve the ecological environment of paddy fields, and enhance soil fertility [21,22]. The published research on the rice–frog co-cultivation system has mainly focused on the impact of frog activities on changes in paddy field soil nutrients, soil microbial diversity, and rice yields. As an assemblage jointly influenced by two very different organisms and multiple environmental factors, rice–frog co-cultivation systems have not been explored by studies beyond their effects on soil nutrient levels [23,24]. Moreover, compared with conventional cultivated land, reclaimed fields have critical disadvantages, such as a weaker soil fertility and poorer production capacity. Therefore, it is urgent to conduct research on the mechanisms and models for improving the productivity of reclaimed fields. However, research in this area is scarce. We hypothesize that rice–frog co-cultivation modulates soil microbial communities and nitrogen cycling, thereby altering greenhouse gas (CH_4_ and N_2_O) emissions compared to conventional rice monocultures, with implications for both ecological and climatic sustainability. The inclusion of frogs in the co-culture model is based on their ecological roles in paddy ecosystems: (1) bioturbation (e.g., movement and foraging) alters the soil structure and oxygen availability, which modulates microbial habitats; (2) excretion provides additional nitrogen substrates, potentially influencing nitrification/denitrification processes; and (3) predation on pests reduces the need for pesticides, avoiding the disruption of soil microbial communities. These roles make frogs a key driver of ecological interactions in the system, which we hypothesized would affect greenhouse gas-related microbial genes. The present study aims to investigate the impact of the rice–frog co-cultivation model on the physicochemical properties of the reclaimed paddy field soil and the dynamic changes in CH_4_ and N_2_O emissions, thereby revealing its mechanism of action in reducing greenhouse gas emissions from paddy fields and providing a solid scientific basis for the refinement and implementation of this ecologically friendly rice cultivation model.

## 2. Materials and Methods

### 2.1. Study Site

The experimental study site was established in Shafan Township, Jinhua City, Zhejiang Province, China (119°29′36″ E, 28°52′42″ N), which has a subtropical monsoon climate. The experimental site is a reclaimed field that was previously idle agricultural land abandoned for approximately 5 years (from 2019 to 2023) and reclaimed in March 2024 using conventional tillage (plowing to 20 cm depth) without additional soil amendment. During the study period (July–October 2024), this region is characterized by hot and humid conditions, consistent with its long-term annual averages: mean rainfall 1309 mm and mean sunshine duration 1810.3 h. These climatic features form the environmental context for greenhouse gas emissions observed in this single growing season.

### 2.2. Experimental Design

Three experimental groups were established based on the stocking density of black-spotted frogs: rice monoculture control group (CG, <50 frogs/mu,1 mu ≈ 666.67 m^2^), low-density rice–frog co-culture (LRF, 5000 frogs/mu), and high-density rice–frog co-culture (HRF, 10,000 frogs/mu). In the experiment, ‘Yongyou 31’ rice (*Oryza sativa*) and black-spotted frogs (*Pelophylax nigromaculatus*) were selected as the co-culture combination. The rice was transplanted manually at a density of 15,000–20,000 hills per mu, with 2 seedlings (derived from 2 grains of rice) per hill. After 15 days of rice transplantation, 1–5 g healthy frog fries were released, and the amount of frog feed applied was approximately 5% of the frog body weight. During the experimental period, no fertilizers or pesticides were applied to any of the experimental fields.

Wooden stakes were driven around the experimental paddy fields, and polyethylene nets were used as fences. The fence was 1–1.5 m high and buried to a depth of approximately 10 cm underground to prevent animals from entering or leaving each plot. Two inlets and outlets were set up in each experimental paddy field. In addition to the fence installed to prevent frogs from escaping, a net was secured over all the experimental plots to prevent birds from preying on juvenile frogs. In this way, frogs were effectively prevented from being preyed on by natural enemies or escaping into other fields, thus avoiding risks associated with invasive species. The water was kept clean and circulated to prevent bacterial infection. The entire experimental study period lasted from July to October of 2024, covering a single continuous growing season of late rice with no interval between experimental phases. All treatments (CG, LRF, HRF) were conducted simultaneously within this period, with data collection (soil sampling, gas monitoring) repeated at fixed intervals.

### 2.3. Sample Collection

Soil samples were collected during four key growth periods of rice, namely the tillering stage, the booting stage, the heading stage, and maturity [25]. The S-shaped five-point sampling method was adopted to collect samples from the rice rhizosphere soil at a depth range of 0–20 cm. Each experimental group (corresponding to a distinct treated plot, as specified in Section 2.2) collected five subsamples from the plot, which were combined, mixed, and analyzed as one composite sample. For each experimental group, 3 replicate composite samples were collected, and with 3 treated plots in total, this resulted in 9 samples (3 treated plots × 3 replicate composite samples) collected during each period. All samples were immediately placed on ice to maintain a temperature below 0 °C after collection and then quickly transported back to the laboratory. After removing impurities from the soil, the samples were mixed evenly. The four-point method was used to divide each sample into two parts; one part was sealed in a sealed bag and stored in a freezer at −80 °C, while the other part was air-dried in a cool and ventilated place. Subsequently, the air-dried samples were ground and screened through a 100-mesh nylon sieve. The screened samples were then placed in labeled sample bags for subsequent determination of various soil physicochemical properties.

Gas samples were regularly analyzed as follows. The CH_4_ and N_2_O emission fluxes during the rice growth period were continuously monitored using the manual static chamber–gas chromatography method [24]. The static chamber consists of a base and a sampling chamber. The dimensions of the chamber are 50 cm × 50 cm × 100 cm, and those of the base are 50 cm × 50 cm × 20 cm. There was a 2 cm deep groove in the base into which the sample chamber was inserted. To ensure proper sealing, a sufficient volume of water was injected into the groove during gas sampling, and the chamber was wrapped with aluminum foil to prevent the temperature inside the chamber from rising too rapidly. Starting one week after the frogs were released, sampling was conducted once every 7 days, between 8:00 and 10:00 am each time. At 0, 5, 10, 15, and 20 min during each sampling, 50 mL of gas inside the chamber was drawn with a 50 mL syringe and put into a vacuum gas sampling bag, which was then brought back to the laboratory for sand determination analysis.

### 2.4. Determination of Key Soil Parameters

Current research suggests that greenhouse gas emissions from paddy fields are affected by the combined action of multiple soil factors, specifically including paddy field soil temperature, soil pH, oxidation–reduction potential (Eh), soil organic matter content (SOM), dissolved organic carbon (DOC) content, cation exchange capacity (CEC), and soil enzyme activity.

The soil Eh was measured using a TDR300 Soil Redox Potential Meter (Sartorius, Bohemia, NY, USA). The soil pH was determined with a PE-10 pH Meter (Sartorius, Göttingen, Germany). The soil dissolved organic carbon (DOC) was measured using a TOC analyzer (Shimadzu, Kyoto, Japan). The soil organic matter (SOM) was determined by the potassium dichromate oxidation (Sigma-Aldrich, St. Louis, MO, USA) method. The CEC of the soil was determined by the hexamine cobalt (III) chloride extraction–spectrophotometric (Alfa Aesar, Ward Hill, MA, USA) method.

#### 2.4.1. Determination of Gas Samples

The collected gas samples were analyzed using a gas chromatograph (GC7890A, Agilent Technologies Inc., Santa Clara, CA, USA). The gas chromatograph was equipped with a flame ionization detector (FID) for the determination and analysis of CH_4_ and equipped with an electron capture detector (ECD) for the determination and analysis of N_2_O [26].

Emission flux was calculated as follows [27]:(1)$$F=\rho \;\times \;\frac{V}{S}\;\times \;\frac{dc}{dt}\times \frac{273}{273+T}.$$

Here, *F* represents the emission fluxes of CH_4_ and N_2_O, in units of mg/(m^2^·h) and μg/(m^2^·h), respectively; ρ is the density of the gas under standard conditions, with units of 0.714 kg/m^3^ and 1.964 kg/m^3^ for CH_4_ and N_2_O, respectively; *V* is the effective volume of the sealed chamber (m^3^); *S* is the area of the base (m^2^); *dc*/*dt* is the rate of change in gas concentration within the sealed chamber per unit time; and *T* is the average temperature inside the sealed chamber.

The cumulative greenhouse gas emissions were calculated as follows:(2)$$\mathrm{Ec}=\sum_{i\;=1}^{n}\frac{\left(F_{i}+F_{i+1}\right)}{2}\;\times \;(t_{i+1}-t_{i})\;\times \;24.$$

Here, Ec is the cumulative emissions of the gas, in units of g·m^−2^ or mg·m^−2^; *n* is the number of observations; *F_i_* is the gas emission flux during the *i*-th sampling; *F_i_*_+1_ represents the gas emission flux during the (*i* + 1)-th sampling, in units of mg·m^−2^·h^−1^; and *t_i_* and *t_i_*_+1_ represent the sampling dates of the *i*-th and (*i* + 1)-th times, respectively.

Global warming potential (GWP) was calculated according to the method recommended by the IPCC; on a 100-year timescale, the emissions of CH_4_ and N_2_O were converted to CO_2_ equivalent emissions by multiplication with 25 and 298, respectively. The sum of the two is the GWP. The calculation formula is as follows:(3)$$\mathrm{GWP}=E_{CH_{4}}\;\times \;25\;+\;E_{N_{2}O}\;\times \;298$$

Here, E_CH4_ and E_N2O_ are the emissions of CH_4_ and N_2_O, respectively.

#### 2.4.2. Determination of Soil Enzyme Activity

Soil urease activity was determined by the phenol–sodium hypochlorite colorimetric method; glucosidase activity was determined by the nitrophenol colorimetric method [26]; dehydrogenase activity was determined according to the method described by Guan Songyin [28]; and fluorescein diacetate (FDA) hydrolase activity was determined following the method of Adam and Duncan [29].

#### 2.4.3. The Determination of the Abundance of Relevant Functional Genes

Soil samples were collected during four key rice growth periods, the tillering stage, the booting stage, the heading stage, and maturity, for the determination of the abundance of functional genes related to methanogens (*mcrA*), methanotrophs (*pmoA*), and denitrifiers (*nirS*, *nirK*, *nosZ*) in the soil [30]. Table 1 shows the primer pairs used for PCR amplification of the relevant genes.

The total DNA of soil microorganisms was extracted using the Fast DNA SPIN Kit for Soil (MP Biomedicals, Santa Ana, CA, USA). The DNA quality was confirmed by 1% agarose gel electrophoresis, and the purity and concentration (based on the A_260_/A_280_ value) of soil DNA samples were determined using a NanoDrop ND-2000 spectrophotometer (NanoDrop Technologies, Wilmington, DE, USA). The A260/A280 ratio was required to be between 1.8 and 2.2 to ensure DNA purity. For concentration, the absorbance at 260 nm was measured, with a conversion factor of 1 absorbance unit = 50 ng/μL for double-stranded DNA. The concentration of all extracted soil DNA samples ranged from 30 to 80 ng/μL, ensuring sufficient template for subsequent qPCR amplification (minimum requirement: ≥20 ng/μL). The A_260_/A_280_ value was required to be between 1.8 and 2.2 to ensure the quality of the sample. Real-time quantitative PCR (CFX96 Touch^TM^; Bio-Rad, Hercules, CA, USA) amplification was performed using the primers of the corresponding functional genes (Table 1). The standards of the corresponding functional fragments were serially diluted. Ultra-pure water was used as the blank control, and three replicates of the soil DNA samples were measured together with the standards. Primer specificity was examined based on the qRT-PCR melting curve. A linear regression equation between the Ct value and the logarithm of the copy number was calculated based on the standards with known copy concentrations. The amplification rate and coefficient of determination (*R*^2^) were calculated (amplification efficiency, 97.3–100.7%; *R*^2^, 0.996–1.000). Based on the obtained linear regression equation and the mass of soil used for DNA extraction, the abundance of functional genes was calculated [31].

### 2.5. Statistical Analysis

All data were analyzed using SPSS 26.0 software (IBM Corp., Armonk, NY, USA). Differences in greenhouse gas emissions, soil physicochemical properties, enzyme activities, and functional gene abundances among the three treatments (CG, LRF, HRF) were tested using one-way analysis of variance (ANOVA). Post hoc comparisons were performed using Duncan’s multiple range test to determine significant differences at the *p* < 0.05 level. Correlation analysis was conducted using Pearson’s correlation coefficient, and regression analysis was performed to explore quantitative relationships between soil factors and greenhouse gas fluxes.

## 3. Results

### 3.1. Methane (CH_4_) Emissions of Paddy Fields Under Rice–Frog Co-Cultivation

The CH_4_ emissions from paddy fields in different experimental groups exhibited similar seasonal emission patterns over the rice growing period (Figure 1). The CH_4_ emission flux of paddy fields ranged from −0.3456 to 34.966 mg/m^2^·h. During the rice growth period, there were two peaks in the CH_4_ emission flux. From the first measurement to the tillering stage of the rice, the CH_4_ emission flux gradually increased, and the first peak appeared during the tillering stage. After tillering, the CH_4_ emission flux in paddy fields first decreased and then gradually increased, reaching its second peak during the heading stage of the rice, and then gradually decreased until the rice reached maturity. Overall, the CH_4_ emission flux among different experimental groups was highest in CG plots, followed by LRF and then HRF plots. Compared with the rice monoculture (i.e., CG), the rice–frog co-cultivation systems (i.e., LRF and HRF) significantly reduced the CH_4_ emission flux in paddy fields.

As shown in Figure 2, the cumulative CH_4_ emissions from paddy fields under the rice–frog co–cultivation were significantly lower than those in the rice monoculture across the different rice growth periods. The cumulative CH_4_ emissions over the different growth stages of the rice also exhibited a certain consistency, with the highest levels in the heading stage, followed by maturity, the tillering stage, and the booting stage, in that order, with the heading stage accounting for the majority of the total emissions. During the entire rice growth period, the average cumulative CH_4_ emissions for CG, LRF, and HRF treatments were 249.89, 162.88, and 129.85 kg/ha, respectively, with significant differences among the three groups. Compared with CG plots, the cumulative CH_4_ emissions in LRF and HRF plots decreased by 34.8% and 48.0%, respectively.

### 3.2. Nitrous Oxide (N_2_O) Emissions of Paddy Fields Under Rice–Frog Co-Cultivation

The N_2_O emission fluxes of paddy fields in each experimental group during the rice growth period exhibited similar seasonal trends in variation (Figure 3). Specifically, the N_2_O emission flux gradually increased in the early stage of the rice growth and reached an obvious peak during the rice heading stage. Subsequently, it gradually decreased until the rice reached maturity. The ranges of variation in N_2_O emission fluxes for CG, LRF, and HRF plots were 10.4–98.2, 13.4–125, and 18.1–178 μg/m^2^·h, respectively. Overall, N_2_O emission fluxes were highest in the HRF, followed by the LRF and CG, indicating that the rice–frog co-cultivation significantly increased N_2_O emissions compared with the rice monoculture.

An analysis of the cumulative N_2_O emissions from paddy fields revealed that across the different growth periods of the rice, the cumulative N_2_O emissions in both of the rice–frog co-cultivation treatments were generally higher than those in the rice monoculture group, indicating that the rice–frog co-cultivation significantly increased the N_2_O emissions from paddy fields (Figure 4). The increase in the cumulative N_2_O emissions in the LRF group compared with the CG group was the greatest when the rice had reached maturity, with an increase of 54.9%. The increase in the cumulative N_2_O emissions in the HRF group compared with the CG group was the greatest during the tillering stage, with an increase of 133.5%. The cumulative N_2_O emissions during different growth stages of rice were highest during the heading stage, followed by maturity, the tillering stage, and the booting stage, in that order, with the cumulative N_2_O emissions during the heading stage accounting for the major part of the total emissions. Throughout the entire rice growth period, the cumulative N_2_O emissions from paddy fields in the LRF and HRF groups were significantly higher than those in the CG group, by 37.1% and 96.3%, respectively.

### 3.3. Global Warming Potential of Paddy Fields Under Rice–Frog Co-Cultivation

The analysis of the global warming potential (GWP) showed that CH_4_ dominated the GWP (93.1–98.2%; Table 2). Therefore, the impact of the rice–frog co-cultivation on the GWP closely corresponds to its impact on CH_4_ emissions. Compared with CG plots, the GWP of both LRF and HRF plots was significantly reduced, by 33.5% and 45.3%, respectively.

### 3.4. Soil Enzyme Activity of Paddy Fields Under Rice–Frog Co-Cultivation

The activities of various soil enzymes at different growth stages of rice were determined. The rice–frog co-cultivation generally enhanced the activities of various soil enzymes, especially in HRF plots. Overall, compared with the rice monoculture, the rice–frog co-cultivation had a significantly increased soil urease activity. Among the two co-cultivation treatments, the HRF treatment enhanced the soil urease activity the most (Figure 5A). Throughout the entire rice development period, compared with the CG treatment, the LRF treatment increased the soil urease activity by approximately 3.6–13.5%, while the HRF treatment, with a high frog density, increased the soil urease activity by approximately 8.9–20.0%. Among the two treatments, the difference in the soil urease activity between the HRF and LRF treatments and that of the CG treatment was the greatest during the heading stage.

Compared with the rice monoculture, the rice–frog co-cultivation significantly increased the soil glucosidase activity (Figure 5B). During the tillering stage, compared with the CG plots, the activities of soil glucosidase in the LRF and HRF plots increased significantly, by 8.7% and 14.0%, respectively. During the booting stage, compared with the CG plots, the activities of soil glucosidase in both the LRF and HRF plots increased significantly, by 6.0% and 16.0%, respectively. During the heading stage, compared with the CG plots, the activities of soil glucosidase in the LRF and HRF plots increased significantly, by 17.0% and 18.1%, respectively. When the rice had reached maturity, compared with the CG plots, the activity of soil glucosidase in the HRF plot increased significantly by 19.9%, while that in the LRF increased but not significantly.

Regarding the activity of soil dehydrogenase, the rice–frog co-cultivation was significantly higher than the rice monoculture, and the enhancing effect of the HRF treatment was more significant than that of the LRF treatment (Figure 5C). The activity of soil dehydrogenase in LRF plots was only significantly increased by 8.7% during the heading stage of the rice compared with CG plots, while during the tillering stage, booting stage, and maturity of rice, although it increased compared with CG plots, the difference did not reach a significant level. The activity of soil dehydrogenase in HRF plots during the tillering stage, booting stage, heading stage, and maturity of rice was significantly increased compared with CG plots, with increases of 10.9%, 4.2%, 9.7%, and 10.3%, respectively.

Overall, the soil FDA hydrolase activity was highest in HRF plots, followed by LRF and then CG plots, and the differences between each of the experimental groups were significant (Figure 5D). During the tillering stage, compared with CG plots, the activities of soil FDA hydrolase in LRF and HRF plots both increased significantly, by 11.6% and 21.7%, respectively. During the booting stage, compared with CG plots, the activities of soil FDA hydrolase in LRF and HRF plots both increased significantly, by 6.2% and 15.5%, respectively. During the heading stage, compared with CG plots, the activities of soil FDA hydrolase in LRF and HRF plots both increased significantly, by 10.5% and 14.7%, respectively. At the rice maturity, compared with CG plots, the activity of soil FDA hydrolase in HRF plots increased significantly, by 6.8%.

### 3.5. Abundance of Functional Genes Related to Greenhouse Gas Emissions in Paddy Fields Under Rice–Frog Co-Cultivation

The abundance of methanogens and methanotrophs was evaluated through the analysis of the abundance of *mcrA* and *pmoA* genes. At different rice growth stages, the experimental groups exhibited the same descending order in their abundance of *mcrA*, which is as follows: CG, LRF, and HRF (Figure 6A). During the tillering stage, booting stage, heading stage, and maturity of rice, compared with the abundance of *mcrA* in CG plots, the abundance of *mcrA* in LRF plots decreased significantly by 53.5%, 50.9%, 58.3%, and 33.5%, respectively. The reduction effect of the HRF treatment was more significant, with decreases of 76.2%, 59.9%, 78.7%, and 40.8%, respectively.

Throughout the growth process of the rice, the abundance of the *pmoA* gene among the different experimental groups was consistently highest in HRF plots, followed by LRF and then CG plots (Figure 6B). During the tillering, booting, and heading stages of the rice, compared with the *pmoA* abundance in CG plots, the *pmoA* abundance in LRF plots increased significantly by 32.7%, 34.5%, and 45.0%, respectively, and the relative increase in HRF plots was more pronounced, with increases of 147.8%, 87.2%, and 97.9%, respectively. At the maturity stage of the rice, the *pmoA* abundance in HRF plots increased significantly, by 32.4%, compared with that in CG plots, while there was no significant difference in the *pmoA* abundance between CG and LRF plots. The *mcrA*/*pmoA* abundance ratio of CG plots was highest, followed by LRF and then HRF plots (Figure 6C). During the tillering, booting, and heading stages, relative to the *mcrA*/*pmoA* gene abundance ratio in CG plots, this ratio in LRF plots decreased significantly, by 65.2%, 63.2%, and 71.6%, respectively, and the decreasing effect was more obvious in HRF plots, with decreases of 90.4%, 78.5%, and 89.4%, respectively. However, when the rice had reached maturity, although the *mcrA*/*pmoA* abundance ratios in LRF and HRF plots decreased compared with those in CG plots, they did not reach a significant level of difference, and there were actually no significant differences among the experimental groups.

During the denitrification of nitrites, nitrite reductases encoded by the *nirS* and *nirK* genes play important roles (Figure 7A). Among the different experimental groups, there were significant differences in the abundance of *nirK* corresponding to the denitrifying bacteria in the soil. During the tillering stage of the rice, the *nirK* abundance in HRF plots was significantly increased, by 47.5% and 36.9%, compared with that in CG and LRF plots, respectively; while the abundance of *nirK* in LRF plots increased compared with that in CG plots, it did not reach a significant level. During the booting stage, heading stage, and maturity stage of the rice, compared with the *nirK* abundance in CG plots, the abundance of *nirK* in LRF plots increased significantly, by 77.8%, 182.3%, and 63.9%, respectively, while that in HRF plots also increased significantly, by 122.6%, 226.9%, and 53.1%, respectively.

There were significant differences in the *nirS* gene abundance, a proxy for the denitrifying bacteria abundance, between the rice–frog co-cultivation treatments and the rice monoculture mode across the different growth stages of rice (Figure 7B). During the tillering stage, the *nirS* abundance in HRF plots was significantly increased, by 98.1%, compared with that in CG plots, while there was no significant difference in its abundance between CG and LRF plots. During the booting stage, heading stage, and maturity stage of the rice, compared with the *nirS* abundance in CG plots, the *nirS* abundance in LRF plots increased significantly, by 72.2%, 53.3%, and 196.7%, respectively, while that in HRF plots also increased significantly, by 89.5%, 129.3%, and 341.7%, respectively.

In the denitrification process, the nitrous oxide reductase protein encoded by the *nosZ* gene plays a crucial role in catalyzing the reduction of N_2_O, which is integral to the entire denitrification process. During the tillering stage of rice, compared with CG plots, the *nosZ* abundance in both LRF and HRF plots decreased significantly, by approximately 42.9% and 22.5%, respectively. During the booting stage, there were no significant differences in the *nosZ* abundance among the experimental groups. Although the abundance of *nosZ* in LRF and HRF plots was lower than that in CG plots, the difference did not reach a significant level. During the heading stage, compared with CG plots, the *nosZ* abundance in LRF and HRF plots decreased significantly, by 32.4% and 71.9%, respectively. At the rice’s maturity, the *nosZ* abundance in LRF plots decreased significantly, by 35.3% compared with CG plots, while there were no significant differences between HRF plots and either CG or LRF plots (Figure 7C).

The combined *nirK* and *nirS* gene abundance to the *nosZ* gene abundance ratio showed a trend of first increasing and then decreasing across the different growth stages of rice. It reached its peak during the heading stage of the rice. Moreover, this abundance ratio under the rice–frog co-cultivation was significantly higher than that under the rice monoculture. This change was consistent with the dynamic variation in the N_2_O emission flux (Figure 7D). During the tillering stage, booting stage, heading stage, and maturity stage of the rice, compared with the ratio of the combined abundance of *nirK* and *nirS* genes and the abundance of *nosZ* alone in CG plots, the ratio in LRF plots increased significantly, by 49.4%, 107.1%, 148.3%, and 284.5%, respectively, while that in HRF plots also increased significantly, by 135.7%, 118.6%, 758.4%, and 335.0%, respectively.

### 3.6. Correlation Analysis Between Soil Factors and Greenhouse Gas Emissions

An analysis of the correlation coefficients between soil factors and CH_4_ emissions from paddy fields was conducted (Figure 8). The methane (CH_4_) emission flux was significantly negatively correlated with the soil pH and cation exchange capacity (CEC) and significantly positively correlated with the *mcrA* abundance as well as the *mcrA*/*pmoA* gene abundance ratio. The cumulative CH_4_ emissions were significantly negatively correlated with the *pmoA* abundance and significantly positively correlated with the *mcrA* gene abundance as well as the *mcrA*/*pmoA* gene abundance ratio.

In addition, the soil pH, cation exchange capacity (CEC), and urease activity were significantly negatively correlated with the *mcrA* abundance as well as the *mcrA*/*pmoA* gene abundance ratio in the soil. This indicated that an increase in these soil factors was associated with a decrease in the abundance of *mcrA*, which is a proxy for methanogens in the soil, thereby reducing the production and emissions of CH_4_. Meanwhile, the activities of soil FDA hydrolase, urease, glucosidase, and dehydrogenase were significantly positively correlated with the *pmoA* gene abundance, which is a proxy for methanotroph abundances. This suggests that an increase in the soil enzyme activity helps to increase the abundance of *pmoA*, which is associated with the abundance of methanotrophs in the soil, thereby promoting the oxidation of CH_4_ and reducing its emissions.

A regression analysis was further conducted to explore the quantitative relationships between various soil factors and the CH_4_ emission flux. The soil pH and cation exchange capacity (CEC) exhibited a significant correlation with the CH_4_ emission flux (*y* = 101 − 16*x*, *R*^2^ = 0.597; *y* = 31 − 2.7*x*, *R*^2^ = 0.467)—that is, as the soil pH and CEC increased, the CH_4_ emission flux gradually decreased (Figure 9A,B). The abundance of *mcrA* and the *mcrA*/*pmoA* gene abundance ratio in soil both exhibited significant positive correlations with the CH_4_ emission flux (*y* = 7 + 5.6 × 10^−7^*x*, *R*^2^ = 0.594; *y* = 10 + 0.72*x*, *R*^2^ = 0.476), indicating that the CH_4_ emission flux increased with both the *mcrA* abundance as well as the *mcrA*/*pmoA* gene abundance ratio (Figure 9H,J). In contrast, the soil redox potential (Eh); the activities of FDA hydrolase, urease, glucosidase, dehydrogenase; and the *pmoA* abundance had a weaker explanatory power for the CH_4_ emission flux and thus could have exerted only a relatively smaller influence (Figure 9C–I).

An analysis of the correlation coefficients between soil factors and N_2_O emissions from paddy fields was also conducted (Figure 10). The nitrous oxide (N_2_O) emission flux was negatively correlated with the *nosZ* abundance in the soil and significantly positively correlated with the soil Eh, urease activity, *nirS* abundance, and the ratio of the combined abundance of *nirK* and *nirS* genes to the *nosZ* gene abundance. The cumulative N_2_O emissions were significantly negatively correlated with the *nosZ* abundance and significantly positively correlated with the soil Eh, urease activity, the *nirS* gene abundance, and the ratio of the combined abundance of *nirK* and *nirS* genes to the *nosZ* gene abundance. Notably, the soil CEC was significantly positively correlated with the abundance of the *nirS* gene and the combined abundance of *nirK* and *nirS* genes to the *nosZ* gene abundance, but it was significantly negatively correlated with the *nosZ* abundance. This indicates that although the soil CEC was not significantly correlated with N_2_O emissions itself, it could be associated with N_2_O emissions through its association with the abundance of functional genes related to N_2_O emissions.

A regression analysis was further conducted to explore the quantitative relationships between various soil factors and the N_2_O emission flux from paddy fields. The soil Eh, the urease activity, the *nirS* gene abundance, and the ratio of the combined abundance of *nirK* and *nirS* genes to the *nosZ* gene abundance exhibited a linear positive correlation with the N_2_O emission flux from paddy fields (*y* = −71 + 0.43*x*, *R*^2^ = 0.452; *y* = −316 + 1.8*x*, *R*^2^ = 0.352; *y* = −22 + 2.5 × 10^−6^*x*, *R*^2^ = 0.648; *y* = 20 + 3.4*x*, *R*^2^ = 0.471). That is, as these factors increased, the N_2_O emission flux from paddy fields also gradually increased (Figure 11B,C,E,G). In contrast, the soil CEC, *nirK* gene abundance, and *nosZ* gene abundance had less explanatory power for the N_2_O emission flux and thus could only have exerted a relatively smaller influence (Figure 11A,D,F).

## 4. Discussion

### 4.1. The Impact of the Rice–Frog Co-Cultivation on CH_4_ Emissions from Reclaimed Paddy Fields

The present study revealed that the CH_4_ emission flux during the rice growth process exhibited two distinct peaks. After the rice transplantation, the CH_4_ emission flux gradually increased, reaching its first peak during the tillering stage. Subsequently, it decreased and then gradually increased again, reaching its second peak during the heading stage. The reason for this pattern may be that during the tillering stage, the rice root system grew vigorously, and the consequent increase in root exudates provided a rich substrate for methanogens [32]. The CH_4_ emission peak during the heading stage may have been caused by an increase in the abundance of methanogens bearing the *mcrA* gene. CH_4_ emissions are the result of the combined effects of two opposing processes, namely CH_4_ production and oxidation, and are influenced by various soil physicochemical properties. Within the environmental context of the subtropical monsoon climate in Jinhua (hot and humid conditions during July–October, as noted in Section 2.1), the present study indicates that the rice–frog co-cultivation significantly affects CH_4_ emissions by altering soil physicochemical properties, the enzyme activity, and the abundance of CH_4_-related functional bacteria. However, due to this study being conducted over a single growing season, these findings may be influenced by the limited seasonal environmental variation. The present study found that compared with the rice monoculture, the rice–frog co-cultivation significantly reduced CH_4_ emissions from paddy fields, and the reduction effect was more significant in the HRF plots. This result is consistent with previous studies on rice–frog [33], rice–shrimp [17], rice–fish [34], and rice–duck [35] systems, indicating that the introduction of aquatic animals helps to reduce CH_4_ emissions from paddy fields.

In rice–frog co-cultivation, the feces of frogs are an abundant source of organic matter, which can significantly increase the contents of the soil organic matter (SOM) and dissolved organic carbon (DOC), providing a rich substrate for methanogens [36] and potentially promoting CH_4_ production. However, the CH_4_ emissions from paddy fields in rice–frog co-cultivation are significantly reduced, which may be owing to the bioturbation of frogs in paddy fields, which changes various properties of the paddy field soil [33]. The present study found that compared with the rice monoculture, rice–frog co-cultivation significantly changed the soil pH, cation exchange capacity (CEC), redox potential (Eh), soil enzyme activity, *mcrA* abundance, *pmoA* abundance, and *mcrA*/*pmoA* gene abundance ratio (*p* < 0.05). The Pearson correlation and regression analyses showed that CH_4_ emissions were negatively correlated with the soil pH, CEC, and pmoA abundance, but they were positively correlated with the mcrA abundance and the mcrA/pmoA gene abundance ratio. These are very likely to be important factors affecting CH_4_ emissions from paddy fields. The frog co-cultivation treatment evaluated in this study had a significant impact on the aforementioned soil physicochemical properties. This may offer a key clue to deciphering the mechanism by which rice–frog co-cultivation reduces CH_4_ emissions relative to rice monocultures.

The soil pH can affect the abundance and activity of methanogens and key enzymes in the CH_4_ production pathway, thereby impacting the production and emission of CH_4_ [37,38]. The present study found that a higher pH in HRF and LRF treatments was associated with a reduced dehydrogenase activity, a proxy for the overall microbial metabolic activity. This suggests that the elevated pH not only reduced the mcrA gene abundance but also suppressed the metabolic activity, further decreasing CH_4_ production. Additionally, the increased FDA hydrolase activity in co-culture treatments, which correlates with the pmoA gene abundance, indicates that frog-induced environmental changes enhanced the methane-oxidizing microbial function, promoting CH_4_ oxidation and reducing emissions. Similar results were also reported by Xu et al. [39]. In addition, in rice–frog co-cultivation, the fertility of the paddy field soil may be significantly increased by both the excrement of frogs and the application of feed [40], resulting in a significant increase in the soil CEC, further reducing the *mcrA* abundance associated with methanogens and decreasing CH_4_ emissions.

In addition, in rice–frog co-cultivation, the activities of frogs in paddy fields increase the contact opportunities between soil and oxygen, thus promoting gas exchange among soil, water bodies, and the atmosphere. This process improves the redox environment of the soil, significantly increasing the soil Eh [16]. This change reduces the efficiency of methanogens in producing CH_4_ in an anaerobic environment, while enhancing the CH_4_ oxidation capacity of methanotrophs in the soil, thereby reducing CH_4_ emissions. Since frog feces and the applied frog feed substantially improve the fertility of the paddy field soil and the activities of frogs in paddy fields observably increase the activities of various enzymes in the soil, the biological environment of aerobic microorganisms in the soil is improved [41]. In the present study, the Pearson correlation coefficient analysis found that the activities of the soil FDA hydrolase, urease, glucosidase, and dehydrogenase were significantly positively correlated with the abundance of the methanotroph *pmoA* gene, further indicating that rice–frog co-cultivation reduces CH_4_ emissions by increasing the soil enzyme activity and the abundance of the methanotrophs harboring the *pmoA* gene.

### 4.2. The Impact of Rice–Frog Co-Cultivation on N_2_O Emissions from Reclaimed Paddy Fields

The formation of N_2_O is extremely complex and affected by various environmental factors [42,43]. The present study found that during the rice growth process, the N_2_O emission flux peaked during the heading stage [44]. This may have occurred because during the heading stage of rice, in order to create a well-aerated environment for the rice roots, intermittent irrigation involving shallow wetting and drying is usually employed to ensure soil aeration, thereby promoting the process of nitrification and increasing the emission flux of N_2_O from paddy fields [45]. N_2_O emissions are mainly driven by soil microbial processes, especially nitrification and denitrification, among which denitrification is the main pathway of N_2_O production in paddy fields [46]. The present study found that, compared with the rice monoculture, the rice–frog co-cultivation significantly increased N_2_O emissions from paddy fields. Compared with the control group (CG), the cumulative emissions of N_2_O from paddy fields in LRF and HRF plots increased significantly—by 37.1% and 96.3%, respectively—and that from HRF plots they also increased significantly, by 43.2%, compared with LRF plots. This result is consistent with previous research on integrative rice cultivation models such as rice–duck [35] and rice–shrimp co-cultivation [47], indicating that the input of aquatic animals can increase the N_2_O emissions from paddy fields.

Under the local climatic conditions (subtropical monsoon, July–October), the increased N_2_O emissions from paddy fields under the rice–frog co-cultivation can be summarized as follows. First, the excrement of frogs in paddy fields supplements the nitrogen levels [48], providing a substrate for nitrification and denitrification reactions, thereby promoting the occurrence of nitrification and denitrification and further driving the emission of the intermediate product N_2_O [35]. Second, the bioturbation of frogs in the field can loosen the surface soil and improve soil permeability [49], thus promoting the emission of N_2_O dissolved in water. In addition, the activities of frogs in paddy fields can not only promote the exchange of soil gases but also promote the growth of the rice, in turn accelerating the oxygen release process of rice roots [24], further promoting the process of nitrification and thus increasing N_2_O emissions [33].

Rice–frog co-cultivation can further impact the N_2_O emissions from paddy fields by altering key environmental factors, such as the soil pH, Eh, and urease activity. Compared with the rice monoculture, the rice–frog co-cultivation significantly increased the soil pH. Changes in the soil pH can affect N_2_O emissions by altering the diversity and abundance of nitrification- and denitrification-associated microbial communities and the nitrogen transformation process [50,51,52]. The present study found that rice–frog co-cultivation increased the soil pH, resulting in increased N_2_O emissions from paddy fields. This result is consistent with the findings of Li et al. [53] and Zhang et al. [54]. Although some studies have found that increasing the soil pH to a near-neutral level (pH > 6.5) can reduce N_2_O emissions, in the present study, rice–frog co-cultivation increased the pH value of acidic soil to a moderately acidic level. In such a case, N_2_O emissions usually increase with the pH [55,56].

The bioturbation of frogs in paddy fields can substantially increase the soil Eh. The Pearson correlation coefficient analysis showed that soil Eh was significantly positively correlated with the N_2_O emission flux; this can be explained by the elevated Eh promoting the nitrification process of nitrifying bacteria, thereby increasing N_2_O emissions [35]. In addition, the soil urease activity directly affects the availability and transformation rate of nitrogen in the soil, thus affecting N_2_O emissions. This sequence of events—frog-induced elevated Eh (aerobic conditions) → increased urease activity (enhanced nitrogen-cycling function) → upregulated *nirS* and *nirK* gene abundance —directly drove higher N_2_O emissions. Specifically, the positive correlation between the urease activity and *nirS* abundance confirms that the enzyme-mediated nitrogen transformation fueled the denitrifier activity. Collectively, these data demonstrate a clear pathway, environmental conditions (Eh) → microbial functional activity (urease) → functional gene expression → N_2_O emissions, with the HRF treatment exhibits the strongest effect due to its more favorable combination of the Eh and enzyme activity. The present study found that the soil urease activity was positively correlated with the N_2_O emission flux, as has been observed in previous research [57]. Rice–frog co-cultivation further increased the N_2_O emissions by increasing the soil urease activity.

Denitrification is an important pathway for the production of N_2_O in paddy fields. Rice–frog co-cultivation significantly increased the abundance of nitrite reductase genes (*nirK*, *nirS*), which are involved in the denitrification process, while the abundance of the gene encoding nitrous oxide reductase (*nosZ*), which reduces N_2_O to N_2_, decreased compared with the rice monoculture. The combined *nirK* and *nirS* to *nosZ* gene abundance ratio can effectively predict N_2_O emissions. The present study found that compared with the rice monoculture, the rice–frog co-cultivation significantly increased this ratio. The Pearson correlation coefficient analysis showed that the N_2_O emission flux was significantly positively correlated with the *nirS* abundance and the combined *nirK* and *nirS* to *nosZ* gene abundance ratio, indicating that rice–frog co-cultivation affects N_2_O emissions by altering the abundance of microbes with functional genes involved in the denitrification process.

In the rice—frog co-cultivation system, the observed 12–15% increase in N_2_O emissions, though seemingly modest, holds significant long-term implications. N_2_O has a global warming potential 298 times that of CO_2_ over a 100-year period and remains in the atmosphere for about 114 years [58]. If this upward trend in emissions persists, especially with the large-scale expansion of the co-cultivation system, it could potentially offset the positive effects of reduced CH_4_ emissions (a 30–40% decrease in our study) and disrupt the regional greenhouse gas balance. Mechanistically, frog activities in the paddy fields, such as soil disturbances, may accelerate the nitrification process (converting NH_4_^+^ to NO_3_^−^), while the alternating wet–dry conditions in paddy soils further favor the denitrification process, thereby increasing the risk of N_2_O release as an intermediate product [59].

To address these concerns, we suggest several management strategies. First, implementing a moist intermittent irrigation regime can effectively reduce the frequency of soil wet–dry alternations, thereby suppressing N_2_O emissions during the denitrification process [60]. Second, applying nitrification inhibitors like dicyandiamide (DCD) or 3,4-dimethylpyrazole phosphate (DMPP) in the base fertilizer can inhibit the activity of ammonia-oxidizing bacteria, reducing N_2_O production during the nitrification stage. Previous research has shown that this approach can decrease N_2_O emissions from paddy fields by 20–35% [61]. Third, based on our research findings, adjusting the stocking density of frogs to 800–1000 individuals per hectare can help balance the soil aeration and nitrogen transformation efficiency, preventing excessive N_2_O emissions caused by over-disturbances.

## 5. Conclusions

This study explored the effects of rice–frog co-cultivation (under different frog densities) on greenhouse gas emissions in reclaimed paddy fields, revealing regulatory mechanisms and implications for sustainable agriculture. Rice–frog co-cultivation significantly reduced CH_4_ emissions (cumulative emissions: CG > LRF > HRF) but increased N_2_O emissions. Since CH_4_ contributed 93.1–98.2% of the global warming potential (GWP), the co-cultivation lowered the overall GWP (LRF: −33.5%; HRF: −45.3% vs. CG), with HRF showing the strongest effect. Mechanistically, the co-cultivation reduced CH_4_ via the increased soil pH, cation exchange capacity (CEC), and *pmoA* gene abundance and the decreased *mcrA* abundance and *mcrA*/*pmoA* ratio. N_2_O increases were linked to a higher CEC, Eh, urease activity, *nirS* abundance, and (*nirK* + *nirS*)/*nosZ* ratio. A limitation is the lack of a non-flooded, frog-free control to distinguish frog-specific effects from those mediated by the soil environment. Future studies incorporating such controls could further resolve direct (e.g., bioturbation) versus soil-mediated impacts on microbial functional genes. The rice–frog co-cultivation shows promise for ecological agriculture. Future research should focus on long-term molecular mechanisms to optimize its application.

## Figures and Tables

**Figure 1 biology-14-00861-f001:**
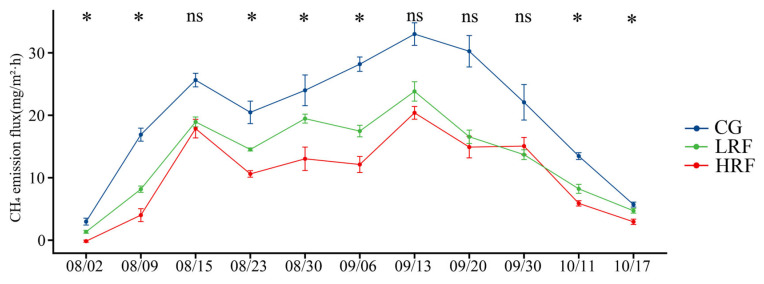
Emission fluxes of CH_4_ under different treatment conditions. Note: For the same time point, * indicates significant differences at the *p* < 0.05 level, and ns indicates no significant difference (one-way ANOVA with Duncan’s post hoc test). CG, rice monocropping group (represented by the blue color description); LRF, low-density rice–frog co-cultivation group (represented by the green color description); and HRF, high-density rice–frog co-cultivation group (represented by the red color description).

**Figure 2 biology-14-00861-f002:**
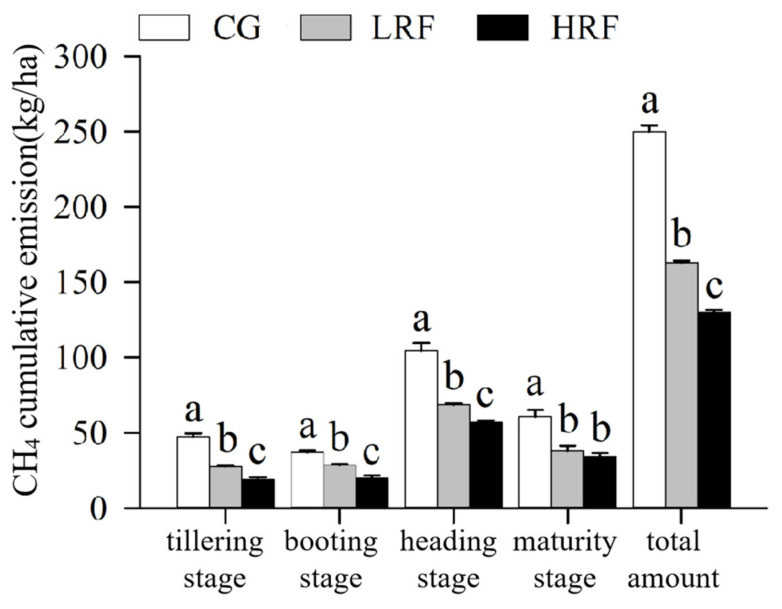
Cumulative CH4 emissions under different treatment conditions across rice growth periods. Note: Significant differences among groups within the same time point are indicated by different lowercase letters (*p* < 0.05, one-way ANOVA with Duncan’s post hoc test). CG, rice monocropping; LRF, low-density rice–frog co-cultivation; and HRF, high-density rice–frog co-cultivation.

**Figure 3 biology-14-00861-f003:**
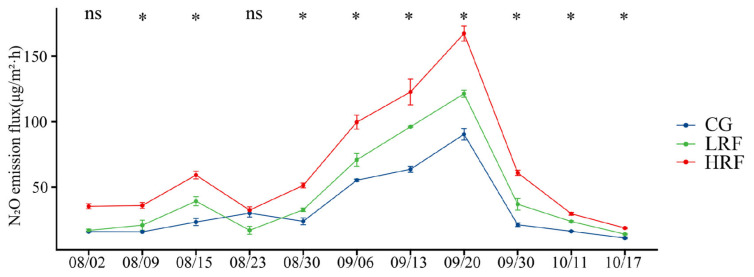
N_2_O emission fluxes of paddy fields under different treatment conditions. Note: For the same time point, * indicates significant differences at the *p* < 0.05 level, and ns indicates no significant difference (one-way ANOVA with Duncan’s post hoc test). CG, rice monoculture (represented by the blue color description); LRF, low-density rice–frog co-cultivation (represented by the green color description); and HRF, high-density rice–frog co-cultivation (represented by the red color description).

**Figure 4 biology-14-00861-f004:**
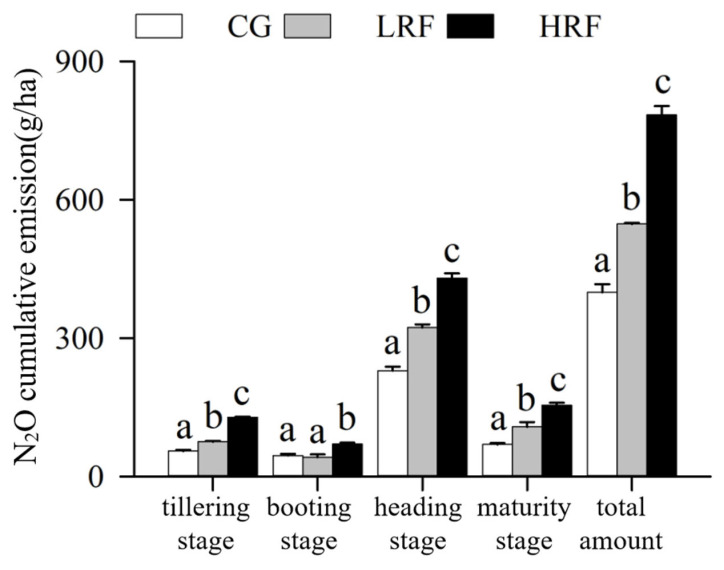
Cumulative N_2_O emissions under different treatment conditions. Note: Significant differences among groups at the same time point are denoted by different lowercase letters (*p* < 0.05, one-way ANOVA with Duncan’s post hoc test). CG, rice monoculture; LRF, low-density rice–frog co-cultivation; and HRF, high-density rice–frog co-cultivation.

**Figure 5 biology-14-00861-f005:**
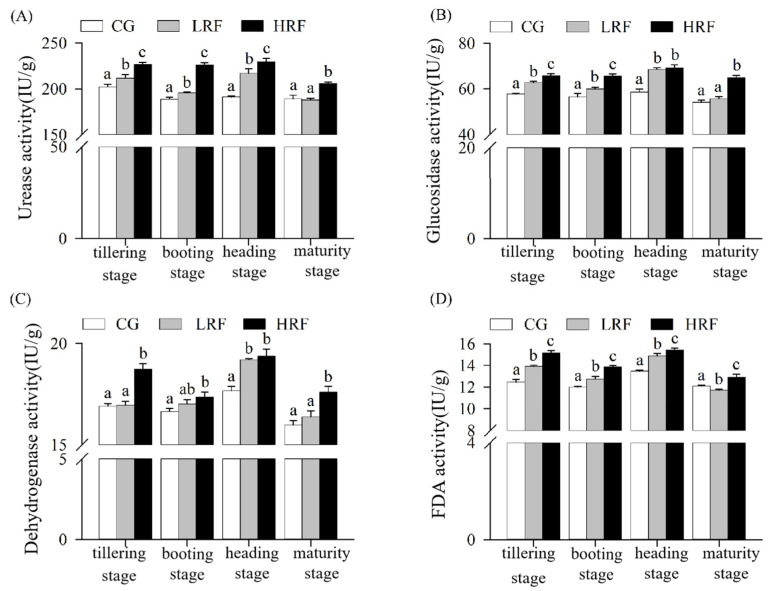
Soil enzyme activities of paddy fields under different treatment conditions. ((**A**), urease activity; (**B**), glucosidase activity; (**C**), dehydrogenase activity; and (**D**), fluorescein diacetate hydrolase activity.) Note: Significant differences among groups within the same sampling period are denoted by different lowercase letters (*p* < 0.05, one-way ANOVA with Duncan’s post hoc test). CG, rice monoculture; LRF, low-density rice–frog co-cultivation; and HRF, high-density rice–frog co-cultivation.

**Figure 6 biology-14-00861-f006:**
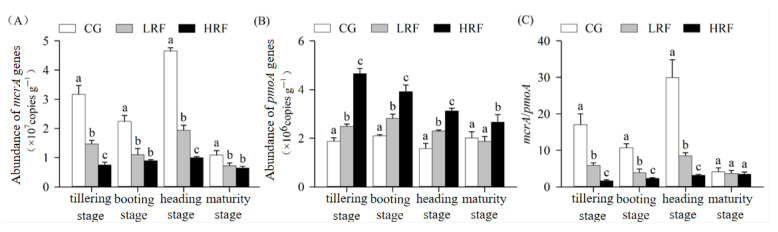
The abundance of CH_4_-related functional genes in paddy fields under different treatment conditions. ((**A**), The *mcrA* gene abundance as a proxy for the methanogenic bacteria abundance; (**B**), the *pmoA* gene abundance as a proxy for the methane-oxidizing bacteria abundance; and (**C**), the *mcrA*/*pmoA* gene abundance ratio as a proxy for the ratio of the methanogenic bacteria abundance to the methane-oxidizing bacteria abundance.) Note: Significant differences among groups within the same time point are indicated by different lowercase letters (*p* < 0.05, one-way ANOVA with Duncan’s post hoc test). CG, rice monoculture; LRF, low-density rice–frog co-cultivation; and HRF, high-density rice–frog co-cultivation.

**Figure 7 biology-14-00861-f007:**
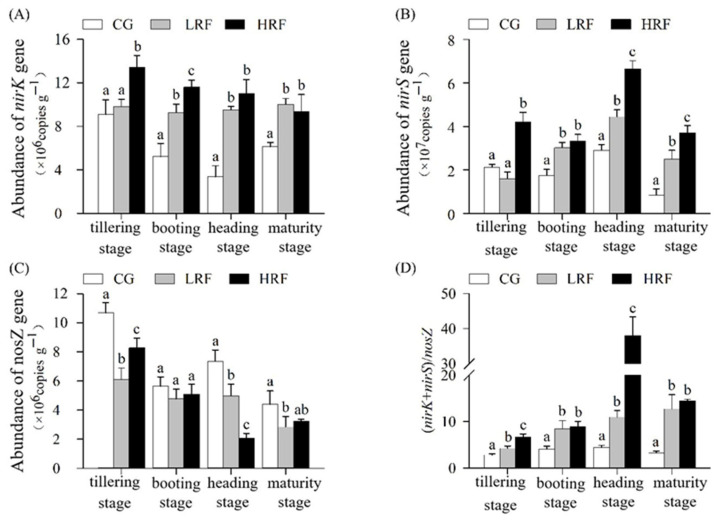
Abundance of N_2_O-related functional genes in paddy fields under different treatment conditions. ((**A**), *nirK* gene abundance; (**B**), *nirS* gene abundance; (**C**), *nosZ* gene abundance; and (**D**), combined *nirK* and *nirS* gene abundance to *nosZ* gene abundance ratio.) Note: Significant differences among groups within same time point are indicated by different lowercase letters (*p* < 0.05, one-way ANOVA with Duncan’s post hoc test). CG, rice monoculture group; LRF, low-density rice–frog co-cultivation; and HRF, high-density rice–frog co-cultivation.

**Figure 8 biology-14-00861-f008:**
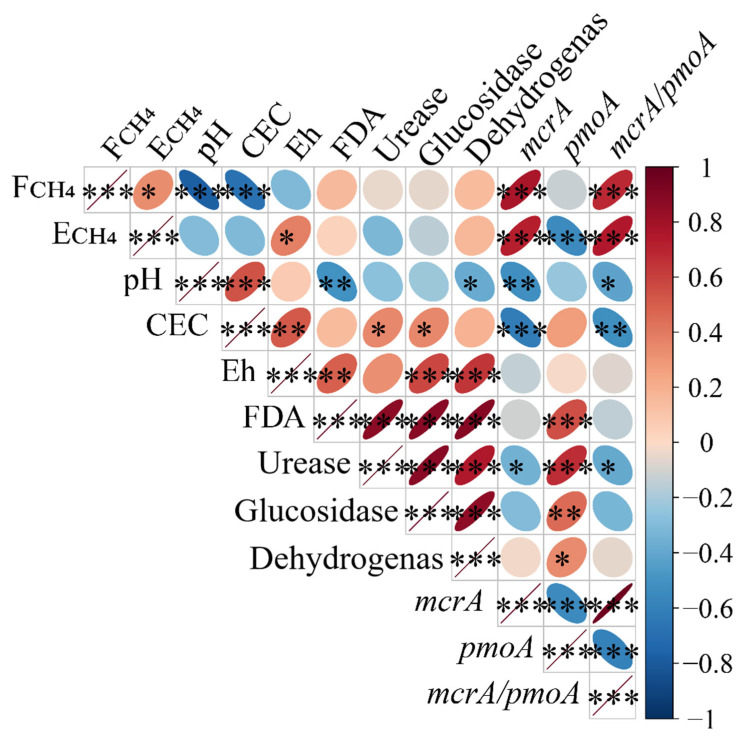
Analysis of correlations between soil factors and CH_4_ emissions. Note: F_CH4_, CH_4_ emission flux; E_CH4_, cumulative CH_4_ emissions; positive correlations are indicated in red, negative correlations are indicated in blue, and narrower ellipses indicate greater correlation coefficients. *, **, and *** indicate significance levels of *p* < 0.05, *p* < 0.01, and *p* < 0.001, respectively.

**Figure 9 biology-14-00861-f009:**
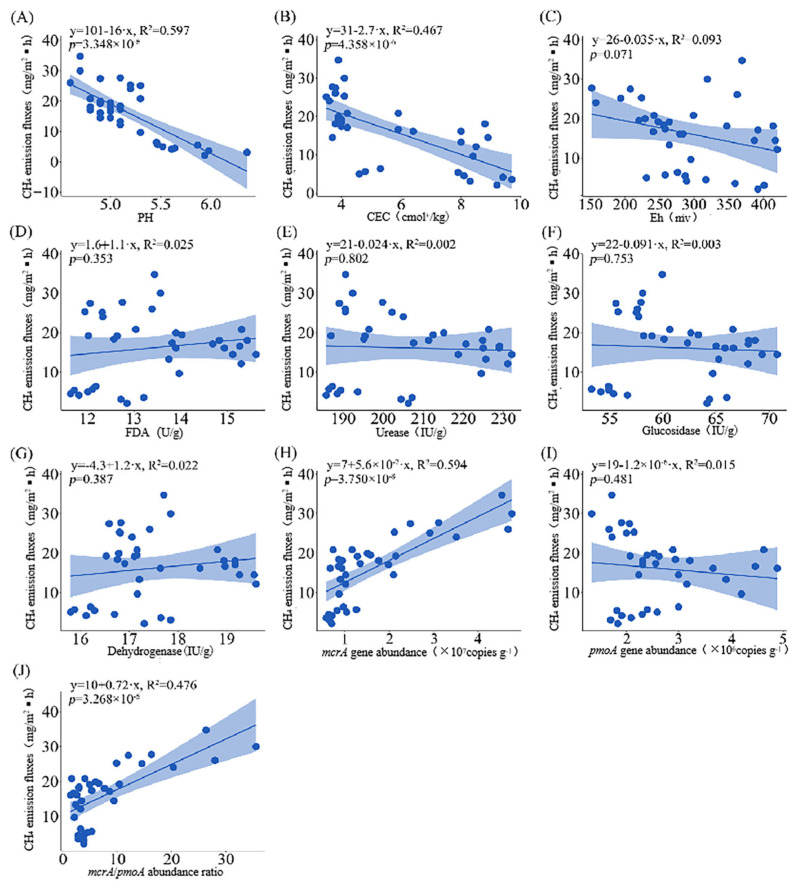
Regression analysis of soil factors and CH_4_ emission fluxes. ((**A**), pH; (**B**), cation exchange capacity; (**C**), oxidation reduction potential; (**D**), fluorescein diacetate hydrolase activity; (**E**), urease activity; (**F**), glucosidase activity; (**G**), dehydrogenase activity; (**H**), *mcrA* gene abundance; (**I**), *pmoA* gene abundance; and (**J**), *mcrA*/*pmoA* gene abundance ratio).

**Figure 10 biology-14-00861-f010:**
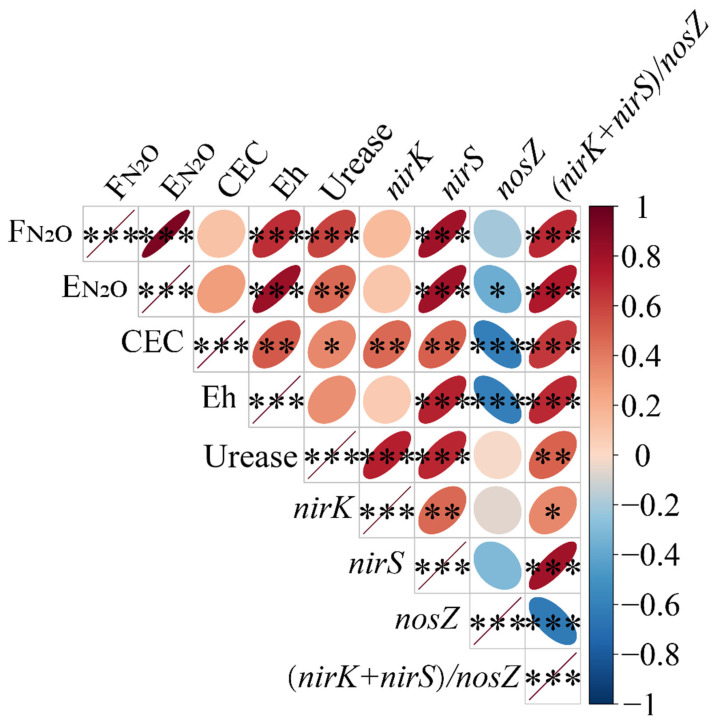
The analysis of the correlation between soil factors and N_2_O emissions. Note: F_N2O_, N_2_O emission flux; E_N2O_, cumulative N_2_O emissions; positive correlations are indicated in red, negative correlations are indicated in blue, and narrower ellipses indicate greater correlation coefficients. *, **, and *** indicate significance levels of *p* < 0.05, *p* < 0.01, and *p* < 0.001, respectively.

**Figure 11 biology-14-00861-f011:**
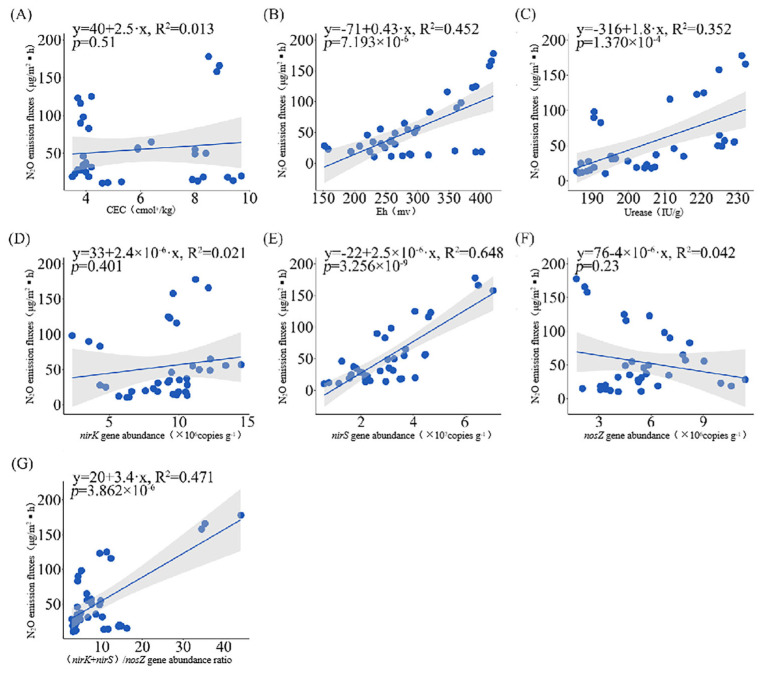
Regression analysis of soil factors with N_2_O emission fluxes. ((**A**), cation exchange capacity; (**B**), oxidation reduction potential; (**C**), urease activity; (**D**), *nirK* gene abundance; (**E**), *nirS* gene abundance; (**F**), *nosZ* gene abundance; and (**G**), combined ratio of *nirK* and *nirS* gene abundance to *nosZ* gene abundance).

**Table 1 biology-14-00861-t001:** Relevant functional gene-specific primer pairs.

Gene	Primer	Sequence
*nirK*	F1aCu	3′-ATCATGGTSCTGCCGCG-5′
	R3Cu	3′-GCCTCGATCAGRTTGTGGTT-5′
*nirS*	nirS-Cd3aF	3′-GTSAACGTSAAGGARACSGG-5′
	nirS-R3cd	3′-GASTTCGGRTGSGTCTTGA-5′
*nosZ*	nosZ-F	3′-AGAACGACCAGCTGATCGACA-5′
	nosZ-R	3′-TCCATGGTGACGCCGTGGTTG-5′
*mcrA*	1106-F	3′-TTWAGTCAGGCAACGAGC-5′
	1378-R	3′-TGTGCAAGGAGCAGGGAC-5′
*pmoA*	A189-F	3′-GGNGACTGGGACTTCTGG-5′
	mb661-R	3′-CCGGMGCAACGTCYTTACC-5′

**Table 2 biology-14-00861-t002:** Global warming potentials under different conditions.

Treatment	CH_4_ Cumulative Emission (kg·ha^−2^)	N_2_O Cumulative Emission (kg·ha^−2^)	Global Warming Potential (kg CO_2_-eq·ha^−2^)
CG	249.89 ± 4.10 a	0.40 ± 0.02 a	6366.30 ± 98.71 a
LRF	162.88 ± 1.34 b	0.55 ± 0.00 b	4235.16 ± 33.12 b
HRF	129.85 ± 1.74 c	0.78 ± 0.02 c	3479.84 ± 44.51 c

Note: Significant differences between groups are indicated by different lowercase letters (*p* < 0.05, one-way ANOVA with Duncan’s post hoc test). CG, LRF, and HRF indicate rice monoculture, low-density rice–frog co-cultivation, and high-density rice–frog co-cultivation, respectively.

## Data Availability

The original contributions presented in this study are included in the article. Further inquiries can be directed to the corresponding author(s).

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
