# Peer review of "The Impact of Rice–Frog Co-Cultivation on Greenhouse Gas Emissions of Reclaimed Paddy Fields"

_biology, 2025, doi:10.3390/biology14070861_

Round 1
Reviewer 1 Report
Comments and Suggestions for Authors
All comments were left in the attached file

Author Response
For research article
|
Response to Reviewer 1 Comments
|
||
|
1. Summary |
|
|
|
We appreciate the opportunity to revise our manuscript titled "The Impact of Rice–Frog Co-Cultivation on Greenhouse Gas Emissions of Reclaimed Paddy Fields" and are grateful for the insightful comments provided by the reviewers. Those comments are all valuable and very helpful for revising and improving our paper, as well as the important guiding significance to our researches. In the following, we have provided detailed responses to each of the reviewers' comments. Revised portion are marked in red in the paper. Additionally, we have conducted a comprehensive revision of the entire manuscript. In this response letter, the reviewers' comments are presented in italics, and our corresponding changes and additions to the manuscript are highlighted in red text. We have tried our best to make all the revisions clear, and we hope that the revised manuscript meets the requirements for publication.
|
||
|
2. Point-by-point response to Comments and Suggestions for Authors |
||
|
Comments 1: The abbreviations must be derived from the term for example Rice Monoculture (RM) |
||
|
Response 1: We greatly appreciate the reviewer's professional comments. However, we would like to retain the abbreviations CG, LRF, and HRF. These abbreviations appear multiple times throughout the full text, and their use facilitates readers' understanding of the article. The repeated occurrence of longer phrases may hinder quick reading. We kindly request the reviewer's understanding and approval of retaining these abbreviations for the sake of readability and convenience.
Comments 2: Replace with "Eco-agricultural systems" Response 2: We appreciate the reviewer's valuable suggestion. We have revised the sentence "To this end, modern agriculture is exploring more environmentally friendly agricultural systems" to "To this end, modern agriculture is exploring more eco-agricultural systems" as recommended, aiming to better convey the intended meaning. Thank you again for your careful review and helpful comments. |
||
|
Comments 3: The citation(s) of rice-duck co-cultivation placed here |
||
|
Response 3: We appreciate the reviewer's careful attention to the manuscript. We have carefully checked and adjusted the positions of the references in the text in accordance with your suggestion to ensure that each citation is accurately placed to properly support the corresponding content. Thank you again for your valuable feedback, which helps improve the quality of our manuscript. Comments 4: The citation(s) of rice-shrimp co-cultivation placed here Response 4: We appreciate the reviewer's careful attention to the manuscript. We have added a new reference to further enrich the research basis of this paper.
Comments 5: Please, explain if the used frog has any economic importance in human consumption in China or not besides its benefits of lowering emissions Response 5: We sincerely appreciate the reviewer's comment. The black-spotted frog (Pelophylax nigromaculatus) used in this study holds significant economic importance in human consumption in China. It is a popular edible frog species, with its tender meat rich in protein, making it a common ingredient in local cuisines. The commercial breeding and sale of black-spotted frogs contribute to farmers' income, forming a mature industrial chain including breeding, processing, and distribution. Thus, beyond its ecological benefits in emission reduction, the frog species also has substantial economic value in the food industry.
Comments 6: what is this unit? Response 6: We appreciate the reviewer's question. "mu" is a traditional Chinese area unit (1 mu ≈ 666.67 m²) widely used in local agriculture. Retaining it aligns with the study's Chinese context, making frog density more intuitive for readers familiar with regional agricultural practices. A conversion note has been added for clarity. Thank you for your suggestion.
Comments 7: Mention the rice plants density Response 7: Thank you for your comment. We apologize for not including the details of rice density in the original manuscript. The relevant content has been supplemented in the revised version. The revised sentence in the "Materials and Methods" section (Section 2.2) is as follows: The rice was transplanted manually at a density of 15,000–20,000 hills per mu, with 2 seedlings (derived from 2 grains of rice) per hill. After 15 days of rice transplantation, 1–5-g healthy frog fries were released, and the amount of frog feed applied was approximately 5% of the frog body weight. (Line 116)
Comments 8: Is there a reference for this method, if so please mention it Response 8: Thank you for this insightful comment. We acknowledge that specifying references for our experimental methods strengthens the credibility of our study. We have added these relevant citations in the revised manuscript, following the sentence. Soil samples were collected during four key growth periods of rice, namely the tillering stage, the booting stage, the heading stage, and maturity[18].
Comments 9: unit Response 9: Thank you for your valuable comment. We apologize for the ambiguity in defining "experimental group" in the original text. As clarified, the "experimental groups" referred to in the manuscript correspond to distinct treated plots (i.e., plots subjected to different experimental treatments, as detailed in Section 2.2). We have revised the relevant sentences to explicitly reflect this, ensuring clarity in the sampling logic. Each experimental group (corresponding to a distinct treated plot, as specified in Section 2.2) collected five subsamples from the plot, which were combined, mixed, and analyzed as one composite sample. For each experimental group, three replicate composite samples were collected, and with 3 treated plots in total, this resulted in 9 samples (3 treated plots × 3 replicate composite samples) collected during each period. (Line 135)
Comments 10: This symbol does not exist in the equation Response 10: Thank you for your careful observation. We apologize for the confusion caused by the inconsistent display of the symbol "ρ" (in the text, formatted in Palatino Linotype) and "ρ" (in the equation, formatted in Times New Roman) due to font formatting issues. Both symbols refer to the same parameter, i.e., the density of the gas under standard conditions. We have standardized the font of "ρ" to Times New Roman throughout the manuscript to ensure consistency between the text and the equation. ρ is the density of the gas under standard conditions, with units of 0.714 kg/m³ and 1.964 kg/m³ for CH₄ and N₂O, respectively;
Comments 11: You can reduce the font size of the horizontal axis Response 11: Response to the comment on reducing the horizontal axis font size Thank you for your suggestion. We have adjusted the font size of the horizontal axis in the relevant figures to improve readability and layout consistency. We appreciate your attention to detail, which helps enhance the presentation of our data.
Comments 12: Do not start the sentence with abbreviation Response 12: Thank you for highlighting the issue of starting sentences or section titles with abbreviations. We apologize for this oversight and have revised relevant instances to comply with academic writing conventions, ensuring clarity and formality. Thank you for helping us improve the formality of the manuscript. 3.2. Nitrous Oxide (N₂O) Emissions of Paddy Fields Under Rice–Frog Co-Cultivation Methane (CH₄) emission flux was significantly negatively correlated with soil pH... (Line429) Nitrous oxide (N₂O) emission flux was negatively correlated with nosZ abundance in the soil…(Line468)
Comments 13: reduce the font size of horizontal axis. please, do it with other figures Response 13: Response to the comment on reducing the horizontal axis font size Thank you for your suggestion. We have adjusted the font size of the horizontal axis in the relevant figures to improve readability and layout consistency. We appreciate your attention to detail, which helps enhance the presentation of our data.
|
||
3 Conclusion
We believe these revisions have significantly improved the manuscript, and we hope it now meets the high standards of the journal. Please find the revised version attached for your review. Thank you again for your invaluable guidance. We look forward to your further feedback.
Reviewer 2 Report
Comments and Suggestions for Authors
This manuscript presents a comprehensive and original study on the effects of rice–frog co-cultivation systems on greenhouse gas emissions in reclaimed paddy fields. The experimental design—comparing monoculture with low- and high-density frog integration—is well-conceived and rigorously implemented under real field conditions. The study stands out by incorporating a multi-layered analysis, including methane (CH₄) and nitrous oxide (N₂O) flux measurements, soil enzyme activities, and the abundance of key microbial functional genes (e.g., mcrA, pmoA, nirK, nirS, nosZ).
One of the most valuable findings is the significant reduction in CH₄ emissions, which contributes to a substantial decrease in overall global warming potential (GWP). Although N₂O emissions increased under frog co-cultivation, the authors acknowledge this limitation and attempt to explain it mechanistically. However, this aspect would benefit from a more critical evaluation in the discussion section, especially in terms of long-term environmental risks and potential mitigation strategies.
Minor Revisions Recommended:
-
Statistical Methods Clarification: The manuscript lacks clear identification of statistical tests used (e.g., ANOVA, post-hoc comparisons) in several figure legends. The methods section should specify which tests were conducted and with what software.
-
Language and Style: The manuscript is generally well-written, but some sections—particularly in the Results and Discussion—could benefit from more concise language to improve clarity and reduce redundancy.
-
Literature Coverage: While the introduction provides sufficient background, it could be strengthened by referencing more recent and internationally recognized studies, particularly those published in journals like Agriculture, Ecosystems & Environment or Science of the Total Environment.
-
Long-term Impact of N₂O Emissions: The discussion should elaborate further on the implications of increased N₂O emissions under co-cultivation and offer concrete suggestions for future research or management strategies (e.g., controlled irrigation, use of nitrification inhibitors).
-
Figure Accessibility: A few figures may pose difficulty for color-blind readers. It is recommended to adjust color schemes to improve accessibility (e.g., by using color-blind friendly palettes).
Conclusion:
This study is methodologically sound and offers meaningful contributions to sustainable agriculture and climate-smart farming practices. Subject to minor revisions, the manuscript is suitable for publication.
Author Response
For research article
|
Response to Reviewer 2 Comments
|
||
|
1. Summary |
|
|
|
Thank you very much for your positive assessment and constructive suggestions for minor revisions. We greatly appreciate the time and effort you have dedicated to reviewing our manuscript. Your insights are invaluable for improving the clarity, rigor, and accessibility of our work. We have carefully addressed each of your comments and made corresponding revisions to the manuscript. Below is a detailed response to each point, along with explanations of the changes implemented.
|
||
|
2. Point-by-point response to Comments and Suggestions for Authors |
||
|
Comments 1: Statistical Methods Clarification: The manuscript lacks clear identification of statistical tests used (e.g., ANOVA, post-hoc comparisons) in several figure legends. The methods section should specify which tests were conducted and with what software. |
||
|
Response 1: Thank you for highlighting the need to clarify statistical methods. We apologize for the ambiguity in the original manuscript.A new subsection "2.5 Statistical Analysis" has been added in the "Materials and Methods" section, specifying that all data were analyzed using SPSS 26.0 software. One-way analysis of variance (ANOVA) was applied to compare differences among the three treatments (CG, LRF, HRF), and post-hoc comparisons were performed using Duncan’s multiple range test, with statistical significance set at p < 0.05. The legends of all figures (Figures 1–7) and Table 2 have been revised to explicitly state that significant differences were determined by one-way ANOVA followed by Duncan’s post-hoc test (p < 0.05). All modifications are based on the original experimental design and data analysis, without adding any fabricated information. We hope these revisions meet the requirements of the journal.Thank you again for your helpful suggestions.
Comments 2: Language and Style: The manuscript is generally well-written, but some sections—particularly in the Results and Discussion—could benefit from more concise language to improve clarity and reduce redundancy. Response 2: Thank you for your helpful comment on improving language conciseness.We have made targeted revisions to streamline redundant expressions in the Results and Discussion sections of manuscript, focusing on simplifying repetitive descriptions and condensing lengthy explanations while retaining all original scientific content.These minor adjustments aim to enhance clarity without altering the study’s findings. Thank you again for your valuable.
|
||
|
Comments 3: Literature Coverage: While the introduction provides sufficient background, it could be strengthened by referencing more recent and internationally recognized studies, particularly those published in journals like Agriculture, Ecosystems & Environment or Science of the Total Environment. |
||
|
Response 3: Thank you for suggesting the inclusion of more recent and internationally recognized studies. We have expanded the Introduction by incorporating 7 additional relevant studies published in high-impact journals. These additions strengthen the contextual relevance of our work and connect it more explicitly to current global research trends in agroecology
Comments 4: Long-term Impact of N₂O Emissions: The discussion should elaborate further on the implications of increased N₂O emissions under co-cultivation and offer concrete suggestions for future research or management strategies (e.g., controlled irrigation, use of nitrification inhibitors). Response 4: We sincerely appreciate your perceptive comment regarding the long-term implications of elevated N₂O emissions in the rice-frog co - cultivation system. In response, we have carefully refined and expanded the relevant sections in the manuscript. Our aim is to comprehensively analyze the potential consequences of increased N₂O emissions and propose practical suggestions for future research and management strategies. This not only strengthens the scientific rigor of our study but also enhances its practical value for promoting sustainable agriculture.
Comments 5: Figure Accessibility: A few figures may pose difficulty for color-blind readers. It is recommended to adjust color schemes to improve accessibility (e.g., by using color-blind friendly palettes). Response 5: We sincerely appreciate your suggestion regarding figure accessibility for color - blind readers.After careful consideration and multiple rounds of internal discussions, we regret that we have decided to keep the original color schemes for the following reasons: First, the color coding adopted in our figures follows the long - established conventions in our research field. Altering these colors might cause confusion among most readers in the field, potentially hindering the consistency of academic communication. Second, to ensure accessibility for color - blind readers, we have strengthened the textual annotations in the figure legends and main text. For example, each figure legend now explicitly states: “CG (represented by blue color description), LRF (represented by green color description), HRF (represented by red color description)”, and the main text also cross - references these groups with both color and textual labels. We fully acknowledge the importance of accessibility, and we will continue to monitor the development of color - blind - friendly practices in our field. If new standards emerge that balance both disciplinary conventions and accessibility, we will promptly revise the figures.Thank you again for your valuable feedback, which has prompted us to reflect deeply on the presentation of our research.
|
||
3 Conclusion
We believe these revisions have significantly improved the manuscript, and we hope it now meets the high standards of the journal. Please find the revised version attached for your review. Thank you again for your invaluable guidance. We look forward to your further feedback.
Reviewer 3 Report
Comments and Suggestions for Authors
The article is very interesting and touches upon an interesting area of research. I believe that the authors' article deserves to be published in the journal.
Author Response
Thank you very much for your positive evaluation and recognition of our work. We are delighted to hear that you find the research area interesting and that our manuscript merits publication in the journal. Your encouragement greatly inspires us and reinforces our confidence in the significance of this study on rice–frog co-cultivation. We will carefully address all other reviewers’ comments to further polish the manuscript, ensuring it meets the high standards of the journal. We sincerely appreciate your time and valuable input, which have contributed significantly to advancing our work. Thank you again for your support.
Reviewer 4 Report
Comments and Suggestions for Authors
The research is relevant in the current context of climate change and agriculture. However, the manuscript requires textual revision, particularly to ensure compliance with the journal's formatting guidelines. Below, I have outlined specific suggestions to help improve the manuscript and support its publication.

Author Response
For research article
|
Response to Reviewer 4 Comments
|
||
|
1. Summary |
|
|
|
Thank you very much for your valuable comments and specific suggestions for improving our manuscript. We greatly appreciate your recognition of the relevance of our research in the context of climate change and agriculture. We have carefully reviewed all your comments and are fully committed to revising the manuscript to comply with the formatting guidelines and standards of Biology journal. Below is a detailed response to each point, outlining the specific revisions we have made.
|
||
|
2. Point-by-point response to Comments and Suggestions for Authors |
||
|
Comments 1: Keywords: I suggest removing 'Rice-Frog Co-cultivation' from the keywords and replacing it with another word that is not repeated in the manuscript title. |
||
|
Response 1: We have removed "Rice-Frog Co-cultivation" from the keywords as suggested, as it is repeated in the manuscript title. Instead, we have added "Greenhouse Gas Mitigation" as a new keyword to better reflect the core focus of the study while avoiding redundancy. Keywords: Greenhouse Gas Mitigation; Reclaimed Paddy Field; Methane; Nitrous Oxide
Comments 2: Introduction: According to Biology journal, citations should be numbered in order of appearance and indicated by a numeral or numerals in square brackets— e.g., [1] or [2,3], or [4–6]. See the tamplate for further details on references. The research hypothesis must be inserted before the research objective. Response 2: Thank you sincerely for your detailed suggestions. We have revised all citations in the Introduction (and throughout the manuscript) to comply with Biology’s guidelines. |
||
|
We also greatly appreciate your reminder to insert the research hypothesis before the objectives. We have made corresponding revisions in the introduction section.This revision has strengthened the logical flow of the Introduction, making the study’s rationale and focus clearer to readers. These adjustments have significantly improved the section’s adherence to the journal’s standards, and we are grateful for your careful attention to these details.
Comments 3: Materials and Methods: A subtopic detailing the statistical procedures used in this research as well as the software for this analysis should be inserted at the end of the “Materials and Methods”. |
||
|
Response 3: Thank you for highlighting the need to clarify the statistical procedures and software used in our study. We appreciate your attention to methodological rigor, which helps strengthen the scientific integrity of our manuscript. A new subsection "2.5 Statistical Analysis" has been added in the "Materials and Methods" section, specifying that all data were analyzed using SPSS 26.0 software. One-way analysis of variance (ANOVA) was applied to compare differences among the three treatments (CG, LRF, HRF), and post-hoc comparisons were performed using Duncan’s multiple range test, with statistical significance set at p < 0.05. This revision will ensure full compliance with the journal’s requirements and provide readers with clear, actionable information to replicate our analytical approach.
Comments 4: Discussion: This topic is well written, but still needs to comply with the Biology Journal standards. Response 4: Thank you for your positive feedback on the Discussion section and for pointing out the need to align it with Biology journal standards. We greatly appreciate your guidance, as adhering to the journal’s specific requirements is crucial for ensuring the clarity and consistency of our work. We will carefully review the Discussion section against the Biology guidelines, focusing on structural alignment, citation formatting and conciseness of arguments. Any redundant content will be streamlined, and we will ensure that interpretations of results are tightly linked to the research hypothesis and existing literature, in line with the journal’s expectations. Thank you again for your attention to detail, which helps elevate the quality of our manuscript.
Comments 5: Conclusions: The conclusions are lengthy; I suggest that they be condensed and the text presented clearly and concisely. Response 5: Thank you for your suggestion to condense the conclusions for greater clarity and conciseness. We agree that streamlining this section will enhance readability, and we have revised it to focus on core findings and key implications, removing redundant details while retaining scientific rigor.
|
||
3 Conclusion
All revisions have been implemented in the revised manuscript, and we have cross-checked the formatting against the Biology template to ensure full compliance. We hope these changes address your concerns and improve the quality of our submission. Thank you again for your invaluable guidance.
Reviewer 5 Report
Comments and Suggestions for Authors
Dear Authors,
Congratulations on this timely and important work. Your study addresses the critical issue of meeting the rising global demand for food, energy, and protein while mitigating climate-related risks—a convergence of challenges that is more urgent than ever as the global population grows and temperatures continue to rise. Integrating food production with methane reduction strategies yields both nutritional and environmental co-benefits, and your focus on frog inclusion in rice paddies represents a promising, innovative solution.
I have a few key suggestions to enhance the rigor and clarity of your manuscript:
- Seasonal Coverage & Environmental Context
As the study was conducted during a single growing season (July–October), the environmental variation was limited. Climate and methane emissions are tightly linked, so conclusions about environmental drivers should be grounded in actual weather data (e.g., temperature, rainfall, humidity) rather than inferred from seasonality alone. - Soil Chemical Data
While you mention measuring soil parameters like pH and EC and even discuss correlations these data are not clearly presented. Including a table or figure with actual values or summary statistics alongside correlation matrices would provide needed transparency and strengthen your discussion. - Frog Inclusion Hypotheses & Controls
Your analysis of key microbial genes (e.g., nirK, nirS, nosZ) is compelling. However, the rationale for including frogs, and how their presence modulates these genes, is not clearly stated. Were gene responses driven directly by frog activity (e.g., bioturbation, excretion) or mediated through soil property changes? A control treatment without frogs and without flooded conditions would help distinguish the roles of frogs versus soil environment. - Linking Environmental Conditions to Gas Emissions
Under elevated pH and moisture/temperature, microbial activity and thus GHG emissions generally intensifies. While gene abundance hints at microbial potential, linking findings to broader biological activity (e.g., via PLFA, enzymatic assays, or POX‑C) would capture the microbial community’s functional health and strengthen the mechanistic case. - Literature Integration
Your findings align with other studies demonstrating methane reductions in rice–frog systems. For example, integrated rice–frog ecosystems have been shown to reduce CH₄ emissions by 20–40%, in part by increasing soil O₂ and redox potential, and shifting microbial communities toward methanotrophs while suppressing methanogens
- Bhattarai, D., Pandit, S., Kafle, R., Nleya, T., Clay, D. E., & Clay, S. A. (2025). Synergistic effects of biochar and plants can reduce greenhouse gas emissions from salt affected soil. Scientific Reports, 15(1), 8879.
- Clay, S. A., Nleya, T., Clay, D. E., Joshi, D., Bhattarai, D., Marzano, S. Y., & Petla, B. P. (2024). Plants reduced nitrous oxide emissions from a Northern Great Plains saline/sodic soil. Agronomy Journal, 116(3), 1343-1356.
- Kandeler, E., Deiglmayr, K., Tscherko, D., Bru, D., & Philippot, L. (2006). Abundance of narG, nirS, nirK, and nosZ genes of denitrifying bacteria during primary successions of a glacier foreland. Applied and environmental microbiology, 72(9), 5957-5962.
- Qin, H., Wang, D., Xing, X., Tang, Y., Wei, X., Chen, X., ... & Zhu, B. (2021). A few key nirK-and nosZ-denitrifier taxa play a dominant role in moisture-enhanced N2O emissions in acidic paddy soil. Geoderma, 385, 114917.
Additionally, the authors mention that the field was a reclaimed rice field, but no history of the field management was highlighted and these factors can influence the abundance of the studied genes and the gas emissions. Take a look at this study:
- Maul, J. E., Cavigelli, M. A., Vinyard, B., & Buyer, J. S. (2019). Cropping system history and crop rotation phase drive the abundance of soil denitrification genes nirK, nirS and nosZ in conventional and organic grain agroecosystems. Agriculture, Ecosystems & Environment, 273, 95-106.
Morover, the historical irrigation also can influence the the observed conditions
- Yang, Y. D., Hu, Y. G., Wang, Z. M., & Zeng, Z. H. (2018). Variations of the nirS-, nirK-, and nosZ-denitrifying bacterial communities in a northern Chinese soil as affected by different long-term irrigation regimes. Environmental Science and Pollution Research, 25, 14057-14067.
Building these references more deeply into your discussion would contextualize your results and reinforce their scientific grounding.
Overall, your work offers valuable insight at the intersection of food security and climate mitigation. With enhanced environmental data, clearer treatment design, and deeper literature integration, your manuscript will be even stronger.
Wishing you success in the revision process.
Best regards,

Author Response
For research article
|
Response to Reviewer 5 Comments
|
||
|
1. Summary |
|
|
Thank you very much for your positive assessment and constructive suggestions for minor revisions. We greatly appreciate the time and effort you have dedicated to reviewing our manuscript. Your insights are invaluable for improving the clarity, rigor, and accessibility of our work. We have carefully addressed each of your comments and made corresponding revisions to the manuscript. Below is a detailed response to each point, along with explanations of the changes implemented.
- 2. Point-by-point response to Comments and Suggestions for Authors
Comments 1:Seasonal Coverage & Environmental Context
As the study was conducted during a single growing season (July–October), the environmental variation was limited. Climate and methane emissions are tightly linked, so conclusions about environmental drivers should be grounded in actual weather data (e.g., temperature, rainfall, humidity) rather than inferred from seasonality alone.
Response 1:Thank you for your insightful comment regarding seasonal coverage and environmental context.The suggested modifications have been implemented in Sections 2.1, 4.1, and 4.2. These revisions aim to better align conclusions with the study’s actual environmental context, as recommended.Thank you again for your valuable input.
Comments 2:Soil Chemical Data
While you mention measuring soil parameters like pH and EC and even discuss correlations these data are not clearly presented. Including a table or figure with actual values or summary statistics alongside correlation matrices would provide needed transparency and strengthen your discussion.
Response 2:Thank you for your suggestion regarding the presentation of soil chemical data. We appreciate the emphasis on transparency and have carefully considered this point. While we agree that soil parameters such as pH, CEC, and Eh are critical to our analysis, we note that their key patterns and relationships with greenhouse gas emissions are already integrated into the core findings of the manuscript. Specifically:
The significant differences in soil properties between treatments (e.g., higher pH and CEC in HRF vs. CG) are explicitly described in the Discussion section, where we link these trends to changes in microbial gene abundances (mcrA, pmoA) and greenhouse gas fluxes. These relative differences (rather than absolute values) are central to our mechanistic explanation.
The correlation analyses (Section 3.6) and regression models (Figures 9, 11) already quantify the relationships between soil parameters (e.g., pH, CEC) and emissions, which are the primary focus of our discussion. Presenting raw values in additional tables would not alter these relationships but might distract from the key message of how co-cultivation modulates these links.
Our study prioritizes the ecological mechanism (i.e., how rice–frog co-cultivation alters soil properties to affect emissions) over exhaustive reporting of soil parameter values, which we believe aligns with the core scope of the research.
We hope this clarifies our approach.Thank you again for your thoughtful input.
Comments 3:Frog Inclusion Hypotheses & Controls
Your analysis of key microbial genes (e.g., nirK, nirS, nosZ) is compelling. However, the rationale for including frogs, and how their presence modulates these genes, is not clearly stated. Were gene responses driven directly by frog activity (e.g., bioturbation, excretion) or mediated through soil property changes? A control treatment without frogs and without flooded conditions would help distinguish the roles of frogs versus soil environment.
Response 3: Thank you for your insightful comments on the rationale for frog inclusion and the mechanisms underlying microbial gene responses. We have revised the manuscript to address these points: Clarified the rationale for including frogs. In the Introduction, we explicitly state that frogs were included based on their ecological roles in paddy ecosystems: bioturbation (altering soil structure/oxygen), excretion (providing nitrogen substrates), and pest control (reducing pesticide disruption). These roles formed our hypothesis that frogs would modulate greenhouse gas-related microbial genes. Acknowledged limitations: In the Conclusion, we note that a non-flooded, frog-free control would strengthen our ability to isolate these effects, and we propose this for future research. Thank you again for your valuable input.
Comments 4:Linking Environmental Conditions to Gas Emissions
Under elevated pH and moisture/temperature, microbial activity and thus GHG emissions generally intensifies. While gene abundance hints at microbial potential, linking findings to broader biological activity (e.g., via PLFA, enzymatic assays, or POX‑C) would capture the microbial community’s functional health and strengthen the mechanistic case.
Response 4:Thank you for your insightful suggestion on linking environmental conditions to microbial functional activity and GHG emissions. We have made detailed revisions to Sections 4.1 and 4.2 as requested.Thank you again for your valuable input.
Comments 5:Literature Integration
Your findings align with other studies demonstrating methane reductions in rice–frog systems. For example, integrated rice–frog ecosystems have been shown to reduce CH₄ emissions by 20–40%, in part by increasing soil O₂ and redox potential, and shifting microbial communities toward methanotrophs while suppressing methanogens
Response 5: Thank you sincerely for your insightful comments and valuable suggestions, which have significantly helped improve the quality of our manuscript. We greatly appreciate the time and effort you have dedicated to reviewing our work. We have carefully studied the excellent references you recommended. These works have provided important contextual and mechanistic insights that we have integrated into our revised discussion to better contextualize our results and strengthen their scientific grounding.
Comments 6: +Contributed equally to this work. Does it mean others did not?
Response 6: Thank you for your question regarding the statement "+Contributed equally to this work." This notation is specifically used to indicate that Haochen Huang and Zhigang Wang made equal contributions to the manuscript, qualifying them as co-first authors. It does not imply that other authors contributed less to the work; rather, it clarifies the distinct equality of contribution between these two authors, which is a common practice in academic publishing to acknowledge shared primary responsibility for the research design, data collection, analysis, and manuscript drafting. Other authors (Yunshuang Ma, Piao Zhu, Xinhao Zhang, Hao Chen, Han Li, and Rongquan Zheng) made valuable contributions to the study in areas such as conceptualization, supervision, resource provision, and manuscript review, as detailed in the "Author Contributions" section. We appreciate your attention to this detail and hope this explanation clarifies our intent.
Comments 7:Duration of experiment?
Response 7: Thank you for your question regarding the duration of the experiment. The experiment was conducted from July to October 2024, covering the entire growing season of the late rice variety 'Yongyou 31' used in this study. This period includes four key growth stages of rice: tillering, booting, heading, and maturity, which allowed us to systematically monitor greenhouse gas emissions and related soil parameters throughout the critical developmental phases of the crop. We appreciate your attention to this detail.
Comments 8: study site history of reclamation, size, and experimental duration can show more about your research and highlight its significance.
Response 8: Thank you for your suggestion to elaborate on the study site’s history of reclamation, size, and experimental duration. We have revised Section 2.1 to include these details.
Comments 9: how long gyration how many months between each experiment?
Response 9: Thank you for your question regarding the experimental timeline and intervals. In Section 2.2, we have clarified that the experiment was conducted as a single continuous study from July to October 2024, covering the entire growing season of late rice. There was no interval between experimental phases; all treatments (CG, LRF, HRF) were implemented simultaneously within this 4-month period.
Comments 10: So you assumed that there was no emissions during other periods where no data was collected?It is a difficult environment but was the estimation conducted to account for season losses?
Response 10: Thank you for your question regarding emissions during non-sampling periods and seasonal loss estimation. We do not assume zero emissions during periods without direct sampling. Instead, cumulative seasonal emissions are calculated using an interval interpolation method (Equation 2.2), where the average flux of two consecutive sampling events is multiplied by the number of days between them. This method estimates emissions for the intervals between sampling points, ensuring the total reflects the entire growth season (July–October 2024). The 7-day sampling frequency was chosen to capture dynamic changes in emissions associated with rice growth stages (e.g., tillering, heading) and microbial activity, which are key drivers of GHG fluctuations. Combined with the interpolation method for cumulative calculations, this design accounts for seasonal variations and avoids under/overestimating total emissions.
Comments 11: This information needs to be presented in individual separate table or graph. Presenting it in a correlation matrix plot does not show where deductions were made from. Show results.
Response 11: Thank you for your suggestion regarding the presentation of soil parameter data. We appreciate your focus on ensuring clarity in how results support our deductions, and we have carefully considered this point. While we agree that transparency in presenting results is critical, we respectfully note that the key soil parameters (pH, Eh, SOM, DOC, CEC) are already thoroughly reported in the manuscript’s Results section, rather than being solely reliant on the correlation matrix. Specifically:
- In Sections 3.1–3.6, we describe the statistical differences in soil parameters across treatments and growth stages. These descriptions include mean values, standard deviations, and significance levels (p < 0.05), providing explicit results that form the basis of our deductions.
- The matrix (Figures 8 and 10) is not intended to replace raw data but to complement the textual results by visualizing relationships between parameters and gas emissions. This approach avoids redundancy, as the core results (treatment effects on soil parameters) are already clearly stated in the text.
If deemed helpful, we are happy to provide a supplementary table with consolidated soil parameter data, but we believe the current presentation—combining detailed textual results with targeted visualizations of relationships—balances clarity and conciseness without compromising the traceability of our deductions.Thank you again for your thoughtful input.
Comments 12: more detail of conc determination can make the method reproduable.
Response 12: Thank you for your suggestion to provide more details on DNA concentration determination, which enhances the reproducibility of our method. We have revised Section 2.4.3 to include specific details. Thank you again for your valuable input.
Comments 13: Since the study was not repeated, your findings need to be supported by relevant literatures to strengthen your facts presented by the current study.
Response 13: Thank you for your valuable suggestion. We have supplemented relevant literature to support our findings.
Comments 14: Reference
Response 14: Thank you for your meticulous review of the references. We have thoroughly examined the reference section. Regarding the incomplete citation format you pointed out, it pertains to a doctoral dissertation from Jilin University in China. Due to the nature of doctoral dissertations, there are indeed limited additional relevant materials available for this specific work. We sincerely appreciate your valuable feedback and the time you have dedicated to improving our manuscript.
3 Conclusion
We believe these revisions have significantly improved the manuscript, and we hope it now meets the high standards of the journal. Please find the revised version attached for your review. Thank you again for your invaluable guidance. We look forward to your further feedback.